# *Alaria esculenta, Ulva lactuca*, and *Palmaria palmata* as Potential Functional Food Ingredients for the Management of Metabolic Syndrome

**DOI:** 10.3390/foods14020284

**Published:** 2025-01-16

**Authors:** Emer Shannon, Maria Hayes

**Affiliations:** Food BioSciences, Teagasc Food Research Centre, Dunsinea Lane, Ashtown, D15 DY05 Dublin, Ireland; emer.shannon@hotmail.com

**Keywords:** seaweed, nutraceuticals, functional foods, metabolic syndrome, polysaccharides, polyphenols, peptides, hypertension, type 2 diabetes, angiotensin converting enzyme (ACE-1), α-amylase, lipase

## Abstract

Hypertension, type 2 diabetes (T2D), and obesity raise an individual’s risk of suffering from diseases associated with metabolic syndrome (MS). In humans, enzymes that play a role in the prevention and development of MS include angiotensin converting enzyme (ACE-1) associated with hypertension, α-amylase associated with T2D, and lipase linked to the development of obesity. Seaweeds are a rich source of bioactives consisting of proteins/peptides, polysaccharides, and lipids. This study examined the potential of seaweed-derived bioactives from *Alaria esculenta, Ulva lactuca*, and *Palmaria palmata* as inhibitors of ACE-1, α-amylase, and lipase. In vitro enzyme inhibitory assays were used to quantify the bioactivity of the seaweed extracts and compare their half-maximal inhibitory (IC_50_) values to recognised positive control enzyme inhibitory drugs captopril© (an ACE-1 inhibitor), acarbose (an α-amylase inhibitor), and orlistat (a lipase inhibitor). Three seaweed extracts displayed enzyme inhibitory activities equal to, or more effective than, the reference positive control drugs. These were *P. palmata* peptides (ACE-1 IC_50_ 94.29 ± 3.07 µg/mL, vs. captopril© 91.83 ± 2.68 µg/mL); *A. esculenta* polyphenol extract (α-amylase IC_50_ 147.04 ± 9.72 µg/mL vs. acarbose 185.67 ± 12.48 µg/mL, and lipase IC_50_ 106.21 ± 6.53 µg/mL vs. orlistat 139.74 ± 9.33 µg/mL); and *U. lactuca* polysaccharide extract (α-amylase IC_50_ 168.06 ± 10.53 µg/mL vs. acarbose 185.67 ± 12.48 µg/mL). Proximate analysis also revealed that all three seaweeds were a good source of protein, fibre, and polyunsaturated essential fatty acids (PUFAs). These findings highlight the potential of these seaweeds in the management of diseases associated with MS and as foods.

## 1. Introduction

Hypertension, T2D, and obesity are three central disorders related to increased risk of developing cardiovascular disease [1], dyslipidaemia [2], micro- and macro-vascular damage [3], and renal complications [4]. The prevalence of hypertension in 39–79 year olds worldwide doubled from 0.65 billion in 1990 to 1.28 billion in 2019 [5]. In 2022, it was estimated that 536.6 million people globally suffered from T2D [6], which is predicted to rise to 700 million (10.9% of the world population) by 2045 [7]. The World Health Organisation (WHO) considers obesity to be one side of the double burden of malnutrition [8]. The prevalence of obesity increased from 4.6% of the world population in 1980 to 14.0% in 2019 [9]. ACE-1, α-amylase, and lipase are three key enzymes associated with these disorders. Within the renin-angiotensin-aldosterone system (RAAS), ACE-1 is involved in the regulation of blood pressure [10,11], where it converts angiotensin-I to the active vasoconstrictor angiotensin II [12]. In the digestive system, α-amylase catalyses the breakdown of starch—the main source of carbohydrate in the human diet—into simple sugars, while lipase catalyses the hydrolysis of triglycerides (fats) into free fatty acids [13], which can then be absorbed by cells. Algal polysaccharides, peptides, and polyphenols can inhibit ACE-1, α-amylase, and lipase in vivo by binding competitively at the active site or non-competitively at an allosteric site, thereby altering the conformation of the active site, preventing the formation of the normal enzyme–substrate complex [14,15]. Other mechanisms have been attributed to zinc and hydrogen bond interactions [16]. The stereochemistry of some polysaccharides, peptides, and polyphenols derived from seaweeds is so similar to the substrates that normally bind with ACE-1, α-amylase, and lipase that the activity of the enzymes is significantly reduced [17,18].

Dietary supplementation with bioactive compounds that inhibit the activity of these enzymes has shown potential in vitro and in vivo in the reduction in blood pressure, blood glucose levels, and excess fat absorption from food [19,20,21,22]. Inhibitory actions by compounds occurs through mimicking of the enzyme’s substrate or by blocking the active site of the enzyme [23,24]. Prescription drugs are widely used to treat hypertension, T2D, and obesity, however, side effects have been reported to include steatorrhea, persistent cough, oedema, pruritus, and valvulopathy [25,26,27].

ACE-1, α-amylase, and lipase inhibition by red, brown, and green seaweeds is known. Cermeño et al. [28] determined the in vitro ACE-1 inhibitory effects of an enzymatically derived protein hydrolysate from the red seaweed *Porphyra dioica*, while Kumagai et al. [29] identified an ACE-1 inhibitory peptide from an enzymatic thermolysin hydrolysate of *Pyropia pseudolinearis*. Saito and Hagino [30] found that peptides fractionated by ion-exchange and gel-filtration from *Pyropia yezoensis* inhibited ACE-1 in vivo in a trial using spontaneously hypertensive rats (SHRs). The same *P. yezoensis* peptides significantly reduced blood pressure in a human study with 64 hypertensive patients (age 51–59 years) when administered at 1.8 g/day for 35 days, with no adverse effects on important clinical parameters [31]. *P. palmata* peptides exhibited ACE-1 inhibitory activity in several in vitro studies [31,32,33,34,35] as well as antihypertensive, renin-inhibitory effects in vivo in a trial with SHRs [36]. Additionally, in silico studies of peptides derived from *U. lactuca* [37] and *A. esculenta* [38] have elucidated their binding mechanism to ACE-1.

Protein hydrolysates have also shown α-amylase inhibitory effects. An example is an enzyme hydrolysate generated from *Porphyra* species identified as having α-amylase inhibitory activity by Admassu et al. [39]. However, polysaccharides and polyphenols are more widely reported as inhibitors of α-amylase and lipase. Previously, Ryu et al. [40] found that polyphenol extracts of the brown seaweeds *Ishige foliacea*, *Ishige okamurae Cladophora wrightiana, Ecklonia cava, Eisenia bicyclis,* and *Asparagopsis taxiformis* (red) had strong anti-diabetic activities and inhibited α-amylase in vitro. Polyphenol extracts from Irish *P. palmata* and *A. esculenta* evaluated by Nwosu et al. [41] were found to inhibit α-amylase in in vitro trials. A sulphated polysaccharide isolated from the red seaweed *Bangia fusco-purpurea* by Jiang et al. [42] inhibited α-amylase in vitro. Polysaccharides and polyphenols extracted from *Ulva ohnoi* [43,44] and *U. lactuca* [20,45] inhibited α-amylase and lipase in vitro, while Lakshmanasenthil et al. [46] determined that the polysaccharide fucoidan was a potent α-amylase inhibitor in silico via grid-based docking and pharmacokinetic analysis.

The aim of this study was to extract polysaccharides (Psac), polyphenols (Pphen), and peptides (Pep) from the seaweeds *A. esculenta, U. lactuca*, and *P. palmata* and screen these extracts for their inhibitory effects against ACE-1, α-amylase, and lipase in vitro. The proximate composition of each seaweed was also determined. Total protein, lipids, carbohydrates, ash, moisture, fibre (soluble and insoluble), and phenolic content (TPC) were quantified. The individual monosaccharide components of algal dietary fibre including L-rhamnose, D-galactose, L-arabinose, D-glucose, D-fructose, and D-mannose were quantified by enzymatic assay. Total and individual fatty acid methyl esters (FAMEs) were evaluated using gas chromatography (GC). This is the first time, to the best of the authors’ knowledge, that all extracts (Pep, Pphen, and Psac) from these particular seaweeds were screened for their ability to inhibit lipase, α-amylase, and ACE-1. Inhibition of these enzymes using these specific fractions is innovative as it enhances the inhibition potential compared to whole seaweeds and additionally makes it easier for the actives to be formulated at physiologically relevant concentrations into final food products without impacting sensory acceptability or compromising bioactivity.

*U. lactuca* polysaccharides, *A. esculenta* polyphenols, and *P. palmaria* peptide extracts had inhibitory activities equal to, or significantly more effective than, the positive control drugs captopril©, acarbose, and orlistat when assessed at pharmacologically relevant concentrations. The findings indicate that polysaccharides, polyphenols, and peptides derived from *A. esculenta*, *U. Lactuca*, and *P. palmata* may have potential applications as anti-hypertensive, anti-diabetic, and anti-obesity ingredients for use in supplements or functional foods, and further research in vivo is warranted to assess the digestibility and bioavailability of these bioactive fractions and nutritional extracts.

## 2. Materials and Methods

### 2.1. Materials

*Alaria esculenta* (Linnaeus) Greville, 1830, *Ulva lactuca* (Linnaeus), 1753, and *Palmaria palmata* (Linnaeus) F. Weber & D. Mohr, 1805 were harvested from the West Coast of Ireland (Summer, 2021) and supplied by SeaLac Ltd., Sligo (Sigma-Aldrich, Bray, Co. Wicklow, Ireland). Whole thalli were air-dried, milled (2 mm diameter), and used for analyses. Following solvent extraction and N_2_ drying, all extracts were freeze-dried (72 h, −20 °C, 0.01 atm, Cuddon FD80 Freeze Dryer, Blenheim, New Zealand) and stored at −80 °C until further use.

### 2.2. Chemicals

All chemicals were of analytical grade and sourced from Sigma-Alrich (now Merck), (Merck, Bray, Co. Wicklow, Ireland), unless otherwise indicated. Milli-Q® water (18 mΩ) (Merck Millipore, Molsheim, France) was used for all experiments. A BioVision K719 Angiotensin I Converting Enzyme Inhibitor Screening Assay Kit supplied by Cambridge Bioscience (Cambridge, UK) was used to determine the ACE-1 inhibition activity. Enzymes used included lipase (EC 3.1.1.3) (20,000 U/mg, porcine pancreatic, Sigma-Aldrich-L0382, Wicklow, Ireland), papain (EC 3.4.22.2) (30,000 U/mg, from *Carica papaya* Merck-Millipore-1.07144, Darmstadt, Germany), and α-amylase (EC 3.2.1.1) (1000 U/mg, porcine pancreatic, Roche-10102814001, Dublin, Ireland). Pharmaceutical inhibitor controls used were captopril© (BioVision K228-100-5, Cambridge BioSciences, UK), acarbose (≥95% Sigma-Aldrich-A8980, Dublin, Ireland), and orlistat (≥98% Sigma-Aldrich-O4139, Wicklow, Ireland).

### 2.3. Methods

#### 2.3.1. Proximate Analysis

The moisture content of dried, whole seaweeds was determined gravimetrically using ISO method 6496:1999 [47]. Ash content was quantified using a muffle furnace (Carbolite Gero Ltd., Hope, UK) at 550 °C for 7 h according to ISO method 171:2007 [48]. Total crude protein content was calculated using a protein analyser (LECO FP628 Corp., St Joseph, MI, USA) based on the Dumas combustion method, according to AOAC method 992.23 [49]. Nitrogen-to-protein conversion factors used were 4.17 for *A. esculenta*, 4.24 for *U. lactuca*, and 3.99 for *P. palmata*, according to Biancarosa et al. [50]. Total carbohydrate content was determined using the Dubois phenol-sulphuric acid method [51]. Soluble and insoluble dietary fibre content was measured using AOAC method 991.43-1994 [52] with a dietary fibre analyser (ANKOM, New York, NY, USA). The total phenolic content of the polyphenol extracts was determined using the Folin and Ciocalteu colorimetric AOAC method 2017.13 [53], compared to phloroglucinol standards, and expressed as microgram phloroglucinol equivalents per gram (µg PE/g). The monosaccharides D-glucose, L-rhamnose, D-galactose, L-arabinose, D-xylose, and D-mannose were measured in polysaccharide extracts using Megazyme Enzymatic Assay Kits (Megazyme International Ltd., Wicklow, Ireland) [54].

#### 2.3.2. Crude Protein Extraction

Dried seaweed (50 g) was suspended in deionised water (1 L), placed in an ultra-sonication bath (1 h, 40 kHz, Branson Ultrasonic 2800, Brookfield, CT, USA), and then stirred (12 h, 100 rpm, 4 °C) according to the method of Galland-Irmouli et al. [55] and Fitzgerald et al. [56]. The seaweed solution was then centrifuged (10,000× *g*, 1 h, 4 °C). The supernatant was removed and stored (4 °C). The pellet was re-suspended in water (200 mL) and subjected to a second extraction (12 h, 100 rpm, 4 °C). Proteins were precipitated by bringing the pooled supernatants to an ammonium sulphate saturation of 80% (12 h, 100 rpm, 4 °C) and then centrifuged (1 h, 10,000× *g*, 4 °C). The ammonium sulphate supernatant was discarded. The pellet was transferred to molecular weight cut-off (MWCO) dialysis tubing (3.5 kDa), tied, suspended in water (5 L), and stirred (24 h, 100 rpm, 4 °C) to remove the residual ammonium sulphate. The water was discarded, fresh water (5 L) was added, and the dialysis tubing was stirred (24 h, 100 rpm, 4 °C). The crude protein extract in the tube was retained and freeze-dried (72 h, −20 °C).

#### 2.3.3. Enzymatic Hydrolysis of Crude Protein

The method previously described by Fitzgerald et al. [56] was used to enzymatically hydrolyse the crude protein extracts. Dried, crude protein powder was suspended in water (15 mg/mL) in a Duran bottle (total volume 10 mL). NaOH (0.01 M) was added to adjust the pH to 6.0. Bottles were heated in a shaker-water-bath (100 rpm, 60 °C). Papain was added (20.7 U/mg crude protein powder) and incubated again (100 rpm, 60 °C, 24 h). Papain was heat-deactivated (95 °C, 10 min). Hydrolysates were frozen and freeze-dried (72 h, −20 °C).

#### 2.3.4. Enrichment of Hydrolysates Using Molecular Weight Cut-Off (MWCO) Filtration

MWCO centrifugal filters (Thermo Scientific, Pierce™ Protein Concentrator PES 88525, 3K MWCO, Waltham, MA, USA) were used to extract the peptide hydrolysate fractions of molecular weight 1 and 3.5 kDa. Whole hydrolysate powders were re-suspended in ultrapure water (0.1 g/mL to a total volume of 100 mL) and transferred to centrifugal filters. The filters were centrifuged (4000× *g*, 60 min). The liquid permeate was pooled and freeze-dried as previously described.

#### 2.3.5. Degree of Hydrolysis

The method previously described by Hoyle and Merritt [57] was used to determine the degree of hydrolysis (DH) of the papain-derived hydrolysates by comparing the ratio percentage of nitrogen soluble in 10% trichloroacetic acid (TCA) (Thermo Scientific, Dublin, Ireland) to the total nitrogen in each sample. Whole hydrolysate powder was suspended in water (50 mg/mL). Aliquots of the whole hydrolysate suspension (300 µL) were added to an equal volume of TCA (20% solution) with a final concentration of 10% TCA. Tubes were vortexed, allowed to sit at room temperature (30 min), then centrifuged (3000× *g*, 20 min). Supernatants were transferred to clean tubes. The nitrogen content of the supernatant was quantified using a LECO FP628 protein analyser as previously described. DH was calculated using the equation:$$Degree\;of\;hydrolysis=\frac{Nitrogen\;soluble\;in\;TCA}{Total\;nitrogen}\;*\;100$$

#### 2.3.6. Polysaccharide Extraction

The method of Dore et al. [58] was used to extract polysaccharides from dried seaweed. A pre-treatment was used to remove the lipids and pigments by suspending dried seaweed (10 g) in acetone (50 mL/g) and stirring (3 h, 200 rpm, room temperature). Acetone was removed by centrifugation (5000× *g*, 30 min, SIGMA Model 4-5L, Darmstadt, Germany). The pellet was dried at room temperature under a fume hood (1 h), then suspended in sodium chloride solution (0.25 M, 100 mL) in a Duran bottle. Sodium hydroxide (0.1 M) was added to adjust the pH to 8.0 (Radiometer PHM93 pH meter, Copenhagen, Denmark). The bottles were incubated (60 °C, 200 rpm, 30 min) in an orbital-shaker water bath. Subtilisin A protease (10 mg) was added (24 h, 200 rpm, 60 °C) to release polysaccharides from the protein–polysaccharide complex within the seaweed. The enzyme was deactivated (95 °C, 200 rpm, 10 min). The contents were allowed to cool to room temperature and then filtered through clean muslin cloth. The filtrate was precipitated with an equal volume of ice-cold acetone on a stir-plate (4 °C, 200 rpm, 30 min). Precipitated polysaccharides were collected by centrifugation (30 min, 10,000× *g*,) and freeze-dried.

#### 2.3.7. Polyphenol Extraction

The method previously described by Lopes et al. [59] was used to extract the polyphenols from dried seaweed. Lipids were first removed by stirring dried seaweed (10 g) with *n*-hexane (50 mL, 1 h, 100 rpm) followed by centrifugation (5 min, 4000× *g*). Lipid extraction of the pellet with *n*-hexane was repeated three times. Acetone:water (7:3 *v*/*v*, 50 mL) was added to the de-fatted pellet, stirred (1 h, 400 rpm), and centrifuged (4000× *g*, 5 min) to extract the polyphenols. The supernatants were pooled and evaporated to dryness in a rotary evaporator (30 °C). Pigments were removed from the crude polyphenol extract by dissolving the evaporated residue in methanol (30 mL), then adsorbing onto double the residue mass of cellulose, followed by drying in a rotary evaporator (30 °C). The cellulose was washed with toluene until the filtrate ran clear to remove pigments. The filtrate was discarded. To release the polyphenols, the washed cellulose was filtered with acetone:water (7:3 *v*/*v*, 50 mL). The filtrate was collected and centrifuged (5 min 4000× *g*). The purified polyphenol extract was evaporated in a rotary evaporator (30 °C) until dry.

#### 2.3.8. Quantification of Fatty Acid Methyl Esters by Gas Chromatography

Thirty seven saturated fatty acids (SFA), monounsaturated fatty acids (MUFA), and polyunsaturated fatty acids (PUFA) were identified and quantified in the *A. esculenta*, *P. palmaria*, and *U. lactuca* lipid fractions by gas chromatography using the method described by Araujo et al. [60] with one modification. Instead of using boron trichloride to derivatise the fatty acids (FAs), a mixture of hydrochloric acid (HCl), toluene, and methanol was used as previously described by Ichihara and Fukubayashi [61].

##### Lipid Extraction and Transesterification

Lipids were extracted from whole, dried seaweeds using ethyl-acetate and ethanol at a ratio of 2:1 (*v*/*v*), according to the method previously described Lin et al. [62]. After drying the extracted lipid fractions under nitrogen, aliquots (20 mg) were transferred in triplicate to clean screw-cap glass tubes to which was added toluene (200 µL), methanol (1.5 mL), and 300 µL of 8% (*w*/*v*) HCl in methanol:water (85:15, *v*/*v*) (final HCl concentration 1.2% (*w*/*v*)) [61]. Tubes were vortexed after the addition of each solvent. To initiate the transesterification reaction, the tubes were capped and placed in a heating block (100 °C, 1. h). After cooling to room temperature, hexane (1 mL) and water (1 mL) were added to each tube and vortexed thoroughly. The upper hexane layer was carefully removed by glass pipette, transferred to a clean Eppendorf tube, then syringe-filtered (Sigma-Aldrich Millex Durapore PVDF 0.22 μm pore) into amber glass GC auto-sampler vials. A calibration curve for thirty-seven different fatty acids (FAs) commonly found in algae was generated using a FA standard mix (Supelco 37 Component Fame Mix TraceCERT®, Merck, Arklow, Ireland).

##### Gas Chromatography Conditions

Separation of FAs was achieved with a gas chromatograph system (Nexis™ GC-2030, Shimadzu, Kyoto, Japan) equipped with a flame ionisation detector and liquid auto-sampler, using a polyethylene glycol capillary column (30 m × 0.32 mm × 0.25 µm, SH-FAMEWAX, Shimadzu, Kyoto, Japan). The carrier gas was hydrogen (8 psi, constant pressure mode), at 250 °C, inlet split/splitless, split ratio 25:1. Injection volume was 1 µL. The oven temperature was held at 50 °C for 0.5 min, ramped to 194 °C at 15 °C/min (held for 4 min), then 4 °C/min to 240 °C and held for 3 min. FA peaks were identified through a comparison of their retention time with those of the standards. Shimadzu LabSolutions data analysis software was used to generate chromatograms, construct thirty-seven FA calibration curves, and quantify the FAs from the sample peak areas.

#### 2.3.9. ACE-1 Inhibition Screening

Psac, Pphen, and Pep seaweed extracts were screened for their inhibitory effect against ACE-1 using a BioVision K719 ACE-1 Inhibitor Screening Kit (BioVision Inc., Waltham, MA, USA). Extracts were compared to captopril©, a pharmaceutical ACE-1 inhibitor. Seaweed extracts [S] were diluted with the assay buffer to concentrations of 62.5, 125, 250, 500, and 1000 µg/mL. Each concentration of [S] and inhibitor control [IC] was pipetted (25 µL) in triplicate to a 96-well UV-transmissible plate. Assay buffer alone (25 µL) was used as an enzyme control [EC] in place of seaweed extract or captopril©. Background control [BC] wells were prepared using assay buffer (25 µL) without the addition of ACE-1. ACE-1 solution (40 µL) was added to [S], [EC], and [IC] wells. The total volume of the wells and the [BC] wells was brought to 200 µL with assay buffer, mixed by pipette, and incubated in darkness (37 °C, 20 min). Substrate solution (50 µL) was added to all wells. The OD (optical density) was read (345 nm) initially, then every 5 min for 1 h at 37 °C in a microplate auto-reader.

Slopes were calculated for [S], [IC], [EC], and [BC] by dividing the ∆OD of each well (OD1 − OD2) by ∆t (time) (t2 − t1) in the linear range. The slope of [BC] was subtracted from the slopes of [S], [IC], [EC], and the relative inhibition of ACE-1 was calculated using the following equation:$$\%\;Relative\;Inhibition\;of\;ACE1=\frac{Slope\left[EC\right]-Slope\left[S\right]\;or\left[IC\right]}{Slope\left[EC\right]}\;*\;100$$

To calculate the IC_50_ values, the percentage relative inhibition was plotted against log_10_ of the concentration of each seaweed extract or captopril©. Antilog function was used to revert log_10_ IC_50_ values to µg/mL.

#### 2.3.10. Lipase Inhibition Screening

The lipase inhibitory activity of the Psac, Pphen, and Pep extracts was determined according to the method of Jaradat et al. [63] at concentrations from 62.5 to 1000 µg/mL using *p*-nitrophenyl butyrate as a lipase substrate. The Psac, Pphen, and Pep extracts were dissolved in 10% dimethylsulphoxide (DMSO). Each extract (200 µL) was added to a test-tube with tris-HCl buffer (700 µL, 2.5 mM, pH 7.4 with 2.5 mM NaCl) and porcine pancreatic lipase (100 µL, 200 U/mL). Tubes were vortexed and incubated (15 min, 25 °C). *p*-nitrophenyl butyrate in acetonitrile (100 µL, 10.45 mg/mL) was added to each tube, then vortexed and incubated (37 °C, 30 min). Orlistat, a pharmaceutical lipase inhibitor, was assayed in the same manner in dilutions from 62.5 to 1000 µg/mL as a positive control. An enzyme control was prepared using 10% DMSO in place of the seaweed extract. A 96-well microplate was used to measure the absorbance of samples (200 µL) immediately at 405 nm. The percentage relative inhibition of lipase was calculated using the equation:$$\%\;Relative\;Inhibition\;of\;lipase=\frac{Abs\;enzyme\;control-Abs\;sample\;or\;inhibitor}{Abs\;enzyme\;control}\;*\;100$$

#### 2.3.11. α-Amylase Inhibition Screening

The inhibition of α-amylase by the Psac, Pphen, and Pep extracts (62.5 to 1000 µg/mL) was quantified using a colorimetric Abcam ab283391 α-Amylase Inhibitor Screening Kit (Abcam PLC, Cambridge, UK). Each seaweed test sample [S] was diluted in assay buffer and added (50 µL) in triplicate to the designated wells of a 96-well microplate. Dilutions (62.5 to 1000 µg/mL) of acarbose (50 µL) were added to separate wells as the inhibitor control [IC]. α-Amylase assay buffer (50 µL) was added to three wells as the enzyme control [EC], and 100 µL of buffer to three wells as the background control [BC]. To initiate the reaction, α-amylase enzyme solution was added to each of the [S], [IC], and [EC] wells and mixed thoroughly. The OD was measured at 405 nm at room temperature in kinetic mode for 25 min. Two time points (t1 and t2) were selected in the linear range of the plot, and the corresponding values for the absorbance (OD1 and OD2) were obtained. The OD of the background control [BC] was subtracted from all readings of the [S], [IC], and [EC] wells. The slope of all wells was calculated by dividing the net ΔOD (OD2 − OD1) over time Δt (t2 − t1). The percentage relative inhibition of α-amylase by the seaweed extracts and the acarbose control was calculated as follows:$$\%\;Relative\;Inhibition\;of\;\alpha -Amylase=\frac{Slope\;of\left[EC\right]-Slope\;of\left[S\right]\;or\left[IC\right]}{Slope\;of\;\left[EC\right]}\;*\;100$$

#### 2.3.12. Statistical Analysis

All analyses were conducted in triplicate and expressed as the means ± standard deviation (SD). One-way ANOVA and Tukey’s HSD post hoc test was used to assess the statistically significant differences between means at the 95% confidence interval (GraphPad Prism 9.1.0, San Diego, CA, USA). Half-maximal inhibitory concentrations were calculated by plotting the percentage inhibition of each enzyme versus the log_10_ of the inhibitor concentration in GraphPad Prism by utilising the nonlinear regression analysis function.

## 3. Results

### 3.1. Compositional Content

The proximate composition of whole seaweeds and their crude protein, polysaccharide, and polyphenol extracts is presented in Table 1. The total protein contents of whole, untreated seaweeds (DW) ranged from 6.61 ± 0.08% in *U*. *lactuca* to 9.95 ± 0.21% in *P*. *palmata* and 10.31 ± 0.13% in *A. esculenta*. The protein content of the water-extracted, ammonium sulphate precipitated crude protein extracts was approximately tenfold greater than that of the whole seaweeds, increasing to 74.34 ± 4.56% (*A. esculenta*), 78.77 ± 3.81% (*U. lactuca*), and 60.52 ± 3.09% (*P. palmata*), indicating successful extraction of the protein fraction.

Moisture content was greatest in *P*. *palmata* (91.36 ± 8.38%) and almost identical in *A. esculenta* (87.79 ± 4.65%) and *U. lactuca* (87.60 ± 2.53), or expressed as dry biomass—8.64 ± 0.38%, 12.21 ± 0.64% and 12.40 ± 0.42%, respectively. *A. esculenta* had the highest ash content (16.10 ± 0.08%), followed by *U. lactuca* (14.57 ± 0.37%), then *P*. *palmata* (10.11 ± 0.08%). The total carbohydrate contents of whole seaweeds increased substantially in the Psac fractions after subtilisin enzymatic extraction from 50.49 ± 2.44% to 86.98 ± 3.66% in *A. esculenta*, from 66.04 ± 3.67% to 90.57 ± 4.28% in *U. Lactuca*, and from 69.77 ± 4.89% to 91.98 ± 4.04% for *P. palmata*.

The total fibre content of whole seaweeds only differed by 2% amongst species, varying from 36.10 ± 1.67% in *A. esculenta* to 36.10 ± 1.67% in *P*. *palmata* and 37.50 ± 1.04% in *U. lactuca*. The total fibre content increased modestly in the Psac extracts compared to the whole seaweed by approximately 2% in *A. esculenta* (to 40.30 ± 2.84%) and 5% in *U. lactuca* (to 42.10 ± 2.23%), but increased by 20% in *P. palmata* to 56.00 ± 3.70%. The insoluble fibre content was approximately four times greater than the soluble fibre content in whole *A. esculenta* and *P*. *palmata.* The exception was whole *U. lactuca*, which had 17.10 ± 0.42% soluble fibre but only 3.30% more insoluble fibre (20.40 ± 1.20%). This soluble:insoluble ratio was completely reversed after extraction of the Psac fractions. Soluble fibre content increased more than eightfold in *P. palmata* from 6.70 ± 0.20% in the whole thallus to 53.60 ± 3.50% in its Psac extract. There was a fivefold increase in *A. esculenta* soluble fibre from 7.40 ± 0.30% in the whole thallus to 38.70 ± 2.30% in its Psac extract, and more than a twofold increase in *U. lactuca* from 17.10 ± 0.42% (whole) to 41.10 ± 2.64% (Psac).

The total phenolic contents were similar in all Pphen extracts, ranging from 69.33 ± 2.94 µg PE/g in *P. palmata* to 75.17 ± 3.14 µg PE/g in *U. lactuca*, and 82.83 ± 4.99 µg PE/g in *A. esculenta*. The total lipid contents were all below 1%, ranging from 0.18 ± 0.02% to 0.27 ± 0.09% and 0.48 ± 0.07 in *P*. *palmata*, *A. esculenta*, and *U. lactuca*, respectively.

Total carbohydrates were quantified by the Dubois phenol-sulphuric acid method. Fibre was quantified by the enzymatic gravimetric method. Results are expressed as percentage of dry weight, with the exception of total phenolics, which were expressed as microgram phloroglucinol equivalents per gram of polyphenol extract (DW). All analyses were conducted in triplicate and expressed as the means (*n* = 3) ± standard deviation. Table 2 details the degree of hydrolysis of the papain-derived protein hydrolysates. *P. palmata* was most effectively hydrolysed (80.62 ± 4.86%), followed by *U. lactuca* (75.06 ± 4.03%), then *A. esculenta* (71.43 ± 2.79%).

The individual monosaccharides that are known to comprise soluble and insoluble algal dietary fibre were quantified in each algal Psac fraction and are shown in Table 3. The total monosaccharide contents were similar in all seaweeds, but slightly higher in *P. palmata* (13.75 g/100 g) than *U. lactuca* (11.74 g/100 g) and *A. esculenta* (11.86 g/100 g). Overall, L-arabinose was the most abundant monosaccharide, ranging from 4.37 ± 0.42 g/100 g in *U. lactuca* to 4.60 ± 0.26 g/100 g (*A. esculenta*) and 7.90 ± 0.62 g/100 g (*P. palmata*). L-rhamnose (from 2.47 ± 0.08 to 4.98 ± 0.09 g/100 g), D-glucose (from 0.12 ± 0.02 to 4.16 ± 0.29 g/100 g), and D-galactose (from 0.45 ± 0.01 to 2.95 ± 0.31 g/100 g) were the next most prevalent. D-fructose was only detected in *U. lactuca* (0.35 ± 0.07 g/100 g). Mannose was not detected in any sample.

### 3.2. Fatty Acid Methyl Ester Content

The FA content of each seaweed’s lipid extract as quantified by gas chromatography is set out in Table 4. The GC chromatogram for the *P. palmata* FA profile is presented in Figure 1, showing the SFA C16:0 palmitic acid and the PUFA C20:5 (n-3) cis-5,8,11,14,17-eicosapentaenoic acid (EPA) as the prominent peaks.

The total FA concentrations ranged from 15.496 mg FA/g lipid extract in *U. lactuca* to 19.372 mg FA/g in *P. palmata* and 22.322 mg FA/g in *A. esculenta*. The total SFAs were most abundant in *P. palmata* (10.228 mg FA/g), followed by 7.347 and 4.092 mg FA/g in *A. esculenta* and *U. lactuca*, respectively. C16:0 palmitic acid was the most prevalent SFA in all seaweeds—*P. palmata* contained 5.346 mg FA/g, *A. esculenta* 4.129 mg FA/g, and *U. lactuca* 1.140 mg. C14:0 myristic acid was the second most common SFA at 2.010 mg FA/g in *A. esculenta*, 1.755 mg FA/g in *P. palmata*, and 0.860 mg FA/g in *U. lactuca*. This was followed by C22:0 behenic acid—1.270 mg FA/g in *U. lactuca* and 0.241 mg FA/g in *P. palmata*, while none was detected in *A. esculenta*. C11:0 undecanoic acid and C24:0 lignoceric acid were only found in *P. palmata* (0.820 and 0.239 mg FA/g, respectively). Only *A. esculenta* and *P. palmata* contained C10:0 capric acid (0.179 and 0.215 mg FA/g) and C13:0 tridecanoic acid (0.166 and 0.091 mg FA/g). C18:0 stearic acid was observed in *A. esculenta*, *U. lactuca*, and *P. palmata* (0.358, 0.300, and 0.440 mg FA/g, respectively), and C21:0 heneicosanoic acid—0.434 mg FA/g in *P. palmata* and 0.140 mg FA/g in *U. lactuca* (none detected in *A. esculenta*). C20:0 arachidic acid was not detected in *U. lactuca*, but *A. esculenta* contained 0.225 mg FA/g, and *P. palmata* 0.211 mg FA/g. Lesser quantities of C15:0 pentadecanoic acid (0.155, 0.130, and 0.130 mg FA/g) and C17:0 heptadecanoic acid (0.125, 0.120, and 0.114 mg FA/g) were observed in *A. esculenta*, *U. lactuca*, and *P. palmata*, respectively. Finally, C23:0 tricosanoic acid was absent from *A. esculenta*, but found in *U. lactuca* (0.140 mg FA/g) and in *P. palmata* (0.191 mg FA/g).

MUFAs were differently dispersed in the three seaweeds, being greatest in *P. palmata* (1.296 mg FA/g), 4.392 mg FA/g in *A. esculenta*, and most prevalent in *U. lactuca* (7.256 mg FA/g). The most abundant MUFA was C18:1 trans-9-elaidic acid and cis C18:1 (n-9) oleic acid (combined and quantified as one peak) at concentrations of 3.242, 1.240, and 0.434 mg FA/g in *P. palmata*, *A. esculenta*, and *U. lactuca*, respectively. This was followed by C16:1 palmitoleic acid (0.571, 2.230, and 0.251 mg FA/g in *P. palmata*, *A. esculenta*, and *U. lactuca*, respectively). C17:1 cis-heptadecenoic acid was found at lower levels in *P. palmata* (0.244 mg FA/g), *A. esculenta* (0.120 mg FA/g), and *U. lactuca* (0.160 mg FA/g). C20:1 (n-9) cis-11-eicosenoic acid was only detected in *U. lactuca* (3.330 mg FA/g), and C22:1 (n-9) erucic acid only in *P. palmata* (0.127 mg FA/g). Other quantified MUFAs were C24:1 nervonic acid (ranging from 0.110 to 0.140 mg FA/g), C14:1 myristoleic acid (from 0.097 to 0.116 mg FA/g), and C15:1 cis-10-pentadecenoic acid (from 0.100 to 0.112 mg FA/g).

Within the PUFAs, key findings include the presence in all three seaweeds of the essential dietary fatty acids, C18:3 (n-3) alpha-linolenic acid (ALA) and C18:2 (n-6) cis-linoleic acid (LNA) as well C22:6 (n-3) cis-4,7,10,13,16,19-docosahexaenoic acid (DHA) and C20:5 (n-3) cis-5,8,11,14,17-eicosapentaenoic acid (EPA), which are required for vital cell membrane and brain functions, and normally have to be converted in the body from ALA and LNA [64].

*A. esculenta* had the greatest total PUFA content (10.583 mg FA/g), followed by *P. palmata* (7.848 mg FA/g) and *U. lactuca* (4.149 mg FA/g). The most abundant PUFA in all three seaweeds was EPA (5.563 mg FA/g) in *P. palmata*, while *U. lactuca* contained 2.277 mg FA/g and *A. esculenta* 1.190 mg FA/g of EPA. C20:4 (n-6) arachidonic acid was the next most abundant (3.449 mg FA/g in *A. esculenta*) but was not present in the other two seaweeds. ALA ranged from 2.009 mg FA/g in *A. esculenta* to 0.401 mg FA/g in *P. palmata* and 0.240 in *U. lactuca*. The LNA content was also the highest in *A. esculenta* (1.507 mg FA/g), followed by 0.650 mg FA/g in *U. lactuca* and 0.597 in *P. palmata*. *U. lactuca* contained the most DHA (1.240 mg FA/g), followed by *P. palmata* (0.354 mg FA/g) and *A. esculenta* (0.115 mg FA/g). C22:2 (n-6) cis-13,16-docasadienoic acid was present from 0.441 mg FA/g in *P. palmata* to 0.377 in *A. esculenta* and 0.260 mg FA/g in *U. lactuca*. C18:3 (n-6) gamma-linolenic acid content was 0.276, 0.122, and 0.120 mg FA/g in *A. esculenta*, *P. palmata,* and *U. lactuca*, respectively. C20:2 (n-6) cis-11,14-eicosadienoic acid and C20:3 (n-3) cis-11,14,17-eicosatrienoic acid were observed in all seaweeds from 0.131 to 0.224 mg FA/g. C20:3 (n-6)–cis-8,11,14-eicosatrienoic acid (dihomo-gamma-linolenic acid) was only detected in *A. esculenta* (0.175 mg FA/g) and *U. lactuca* (0.117 mg FA/g). C18:2 (n-6) trans-linolelaidic acid was only found in *U. lactuca* (0.100 mg FA/g).

The summation of the total omega-6 (n-6) and total omega (n-3) FAs in each seaweed is also shown in Table 4. The ratio of n-3 to n-6 FAs in foods is considered an important nutritional parameter as they include the essential dietary PUFA, LNA, and ALA. Within the polyunsaturated FAs, the total n-6 was highest in *P. palmata*, (5.508 mg FA/g), while *A. esculenta* (1.010 mg FA/g) and *U. lactuca* (1.154 mg FA/g) had very similar contents. The total n-3 FAs were differently dispersed, ranging from 3.139 mg FA/g in *U. lactuca* to 5.075 mg FA/g in *A. esculenta* and 6.694 mg FA/g in *P. palmata*. In the case of *A. esculenta*, the n-6 to n-3 ratio was almost equal (1.10:1.02), while *U. lactuca* had approximately three times more n-3 than n-6 (1.01:3.13). The n-3 content of *P. palmata* was more than sixfold greater than its n-6 content (1.15:6.70).

### 3.3. Inhibitory Activities

#### 3.3.1. ACE-1 Percentage Inhibition by Peptide Hydrolysates

Figure 2 shows the percentage relative inhibition of ACE-1 by 1 kDa or 3.5 kDa peptide hydrolysate permeates from each seaweed at concentrations of 0.25, 0.5, and 1.0 mg/mL. Inhibition by captopril© at the same concentrations is also shown.

For all seaweed permeate fractions examined at different concentrations (0.25, 0.5, and 1.0 mg/mL), the 1 kDa MWCO fractions were more effective than the 3.5 kDa fractions at inhibiting ACE-1. The inhibition of ACE-1 by the 3.5 kDa fractions ranged from 67.46 ± 2.10% for *A. esculenta* tested at 0.25 mg/mL to 93.32 ± 6.42% for *P. palmata* tested at a concentration of 1 mg/mL. The 1 kDa fractions exerted greater inhibition, with the lowest also being *A. esculenta* at 0.25 mg/mL (78.35 ± 6.04%) and the greatest *P. palmata* at 1 mg/mL (98.46 ± 5.05%). The captopril© control ranged from 96.60 ± 2.40% (0.25 mg/mL) to 97.87 ± 1.98% (0. 5 mg/mL) and 99.85 ± 1.34% (1.0 mg/mL).

Since the 1 kDa fractions were found to be more effective, these were used to conduct further ACE-1 inhibitory assays at concentrations of 62.5, 125, 250, 500, and 1000 µg/mL to determine the half-maximal inhibitory concentrations.

#### 3.3.2. ACE-1, a-Amylase, and Lipase Half-Maximal Inhibitory Concentrations

Table 5 shows the half-maximal inhibitory concentrations of each seaweed hydrolysate and the permeate fractions against ACE-1, α-amylase, or lipase assayed at concentrations of 62.5, 125, 250, 500, and 1000 µg/mL. Lower IC_50_ values indicate a more effective enzyme inhibitory activity.

ACE-1 inhibition ranged from 94.29 ± 3.07 µg/mL for the *P. palmata* peptide extract to the least effective at 704.69 ± 20.72 µg/mL for the *A. esculenta* polysaccharide fraction. α-Amylase inhibition ranged from 147.04 ± 9.72 µg/mL for *A. esculenta* Pphen to 441.32 ± 23.51 µg/mL for the *U. lactuca* peptide extract. Lipase was also most effectively inhibited by the *A. esculenta* Pphen fraction (106.21 ± 6.53 µg/mL) and least inhibited by the A. esculenta peptide fraction (653.31 ± 36.44 µg/mL).

The most significant results were those from three fractions—the *P. palmata* peptide extract, *A. esculenta* polyphenol extract (Pphen), and *U. lactuca* polysaccharide extract. These extracts had inhibitory effects equal to, or lower than, all of the positive controls. The ACE-1 IC_50_ value of *P. palmata* Pep was 94.29 ± 3.07 µg/mL, which was not significantly different (*p* ≤ 0.05) than that of captopril© (IC_50_ 91.83 ± 2.68 µg/mL). *A. esculenta* Pphen had an α-amylase IC_50_ value of 147.04 ± 9.72 µg/mL and a lipase IC_50_ value of 106.21 ± 6.53 µg/mL, which were both significantly lower than the respective acarbose (185.67 ± 12.48 µg/mL) and orlistat (139.74 ± 9.33 µg/mL) controls. *U. lactuca* Psac had an α-amylase IC_50_ value of 168.06 ± 10.53 µg/mL, which was significantly lower than that of acarbose.

## 4. Discussion

### 4.1. Outcome of Compositional Analysis

The present study highlights the potential health benefits of the whole biomass and polysaccharide, peptide, and polyphenol extracts from the seaweeds *A. esculenta, U. Lactuca,* and *P. palmata.* The selected seaweeds are considered to contain all essential amino acids [65]. The total protein contents of whole seaweeds in the present study ranged from 6.61 ± 0.08% for *U*. *lactuca* to 9.95 ± 0.21% for *P*. *palmata* and were the greatest for *A. esculenta* (10.31 ± 0.13%). Similarly, Stévant et al. [66] found that *A. esculenta* harvested in May from Brittany, France had a total protein content of 10.50%. In a study of Norwegian biomass harvested during May and June, Mæhre et al. [67] reported *A. esculenta* as having 9.11% protein, and *U. lactuca* and *P. palmata* contained 8.65% and 12.26% total protein, respectively. *U. lactuca* harvested in Galicia, Spain (June) had a protein content of 4.65 ± 0.09% [68] while Canadian *P. palmata* harvested from the Gulf of Saint Lawrence in June contained 19.87 ± 0.85% protein [35]. *P. palmata* harvested in different areas of the Gaspe coast, Canada, by Beaulieu et al. [34] had protein contents closer to that of the present study—ranging from 11.07 ± 0.13 to 13.39 ± 0.51% in June and October, respectively. Seasonal variations in the protein content of Scottish *A. esculenta* have also been described by Schiener et al. [69], and quantities ranged from 9.4 ± 0.1% in July, increasing to 11.6 ± 0.3% (May) and 12.1 ± 0.7% (March). Varying protein contents may occur in identical species as extraction can be inhibited by the polysaccharide matrix of the seaweed cell wall and the method of extraction used [70,71].

Moisture content was greatest in *P*. *palmata* (91.36 ± 8.38%), followed by *A. esculenta* (87.79 ± 4.65%) and *U. lactuca* (87.60 ± 2.53), equal to 8.64 ± 0.38%, 12.21 ± 0.64%, and 12.40 ± 0.42% DW, respectively. Contents in a similar range amongst these species were described previously, for example, 85.5 ± 2.5% [69] and 17.2 ± 0.8% (DW) [66] for *A. esculenta*, from 11.6 ± 0.1% (spring) to 16.6 ± 0.02% (winter) (DW) for *U. lactuca* [72], and 8.20 ± 0.77% (DW) for *P*. *palmata* [67]. Ash content ranged from 16.10 ± 0.08% in *A. esculenta* followed by *U. lactuca* (14.57 ± 0.37%), and was lowest in *P*. *palmata* (10.11 ± 0.08%). Varying, but generally higher, ash contents have been reported in all three seaweeds such as 24.2 ± 1.4% and 24.56 ± 0.56% for *A. esculenta* by Stévant et al. [66] and Mæhre et al. [67], respectively. *U. lactuca* harvested in Egypt had an ash content that ranged from 21.5 ± 0.2% in autumn to 28.9 ± 0.3% in winter, according to Mohy El-Din [72], while *P*. *palmata* harvested in northern Spain by Martínez and Rico [73] had the lowest ash content (14%) in autumn, but this increased to 36% in spring. It has been suggested that thallus ash and moisture contents are correlated with light exposure, increasing during the growth phase of the seaweeds (which varies by geographical location and climate), with an accumulation in the growing tips compared to the basal portions [73].

The total carbohydrate contents of whole seaweeds increased substantially in the Psac fractions after subtilisin enzymatic extraction from 50.49 ± 2.44% to 86.98 ± 3.66% for *A. esculenta*, from 66.04 ± 3.67% to 90.57 ± 4.28% (*U. lactuca*), and from 69.77 ± 4.89% to 91.98 ± 4.04% (*P*. *palmata*). In general, slightly lower total carbohydrate contents appear in the literature for *A. esculenta* (40.7 ± 1.5% [66]), *U. lactuca* (24.95 ± 2.11% [74] and 51.0% [75]), and *P*. *palmata* (42–64% [76]). The total fibre content of whole seaweeds increased in the Psac extracts compared to the whole seaweed by 2% in *A. esculenta* (to 40.30 ± 2.84%), 5% in *U. lactuca* (to 42.10 ± 2.23%), and by 20% in *P. palmaria* to 56.00 ± 3.70%. The soluble:insoluble fibre ratio was completely reversed after the extraction of the Psac fractions from the whole seaweed. Soluble fibre content increased in *P. palmata* from 6.70 ± 0.20% (WH) to 53.60 ± 3.50% (Psac), from 7.40 ± 0.30% (WH) to 38.70 ± 2.30% (Psac) in *A. esculenta*, and from 17.10 ± 0.42% (WH) to 41.10 ± 2.64% (Psac) in *U. lactuca*. This increase in soluble fibre in the Psac fractions is most likely due to the degradation of proteins in the seaweed cell wall protein–polysaccharide complex by subtilisin protease extraction and release of the polysaccharides. The complex polysaccharides that comprise each class of seaweed are alginate, fucoidan, and laminarin (brown); ulvan, sulphated-rhamnans, -arabinogalactans, and -mannans (green); and carrageenan, agaran, porphyran, hypnean, and floridean starch (red) [77,78]. These polysaccharides are considered as dietary fibre as they are indigestible within the human stomach and small intestine [79] due to the absence of the necessary fibre-degrading enzymes [80,81]. These have been reported to exert an inhibitory effect on several enzymes in vitro and in vivo including ACE-1, α-amylase, and lipase [20,22,42,82,83,84,85,86,87], which is further detailed in Table 6. An increase in dietary fibre consumption also reduces the biological indices of insulin resistance [88]. Seaweed polysaccharides such as alginate have been successfully incorporated into foods, for example, bread, as nutraceutical ingredients to inhibit lipase [89]. The bread (4% alginate *w*/*v*) retained its lipase inhibitory properties after baking and did not cause any adverse gastrointestinal side effects in a human trial by Houghton et al. [90]. Moroney et al. [91] added an *L. digitata* extract containing 9.3% laminarin and 7.8% fucoidan to minced pork patties (0.1–0.5% *w*/*w*) to increase the fibre content. Ulvan was developed into a colloidal formulation by Morelli et al. [92] for use as a food and drink stabilising and emulsifying agent. Yu et al. [93] used an agar-type galactan from red seaweed to prepare an edible bioactive food film (1.5% agar *w*/*v*) with anti-glycation and other nutraceutical properties.

The individual monosaccharide subunits of these fibres, L-arabinose, L-rhamnose, D-galactose, D-glucose, D-mannose, and D-fructose, were quantified in each Psac fraction (Table 3). L-Arabinose was the most abundant monosaccharide in all three seaweeds, particularly *P. palmata*, where it forms part of the sulphated galactans, carrageenan and agar [94,95], with lower ratios in ulvan [96,97] and fucoidan [98]. Rhamnose was most prevalent in *U. lactuca* as it forms the backbone ulvan—the principle polysaccharide of green seaweeds. As expected, D-glucose was dominant in *A. esculenta* as the primary monosaccharide of laminarin. It was expected that mannose may also be present in *A. esculenta* as it can occur as a side chain of alginate, however, it was not detected. This may be due to quantities being below the limit of detection or sample loss during the hydrolysis preparation step [99]. D-Galactose content was greatest for *P. palmata*, where it occurs as a disaccharide component of floridoside—the main storage carbohydrate in red seaweeds [100]. D-Fructose has been reported to occur in some green species [101], and was present at very low levels in *U. lactuca* only. Broadly varying monosaccharide contents have previously been reported in the literature for *A. esculenta* (8.5 g/100 g glucose [66]); *U. lactuca* (2.22 g/100 g arabinose, 1.01 g/100 g galactose [102], and 5.1 g/100 g rhamnose [103]); and *P. palmata* (3.6 g/100 g glucose [69]).

The total phenolic contents were similar in all Pphen extracts, ranging from the lowest in *P. palmata* (69.33 ± 2.94 µg PE/g) to 75.17 ± 3.14 µg PE/g in *U. lactuca* and 82.83 ± 4.99 µg PE/g in *A. esculenta*. The phenolic compounds found in red and green seaweeds are flavonoids, bromophenols, coumarins, and phenolic acids [104,105,106]. The greater phenol content in *A. esculenta* is most likely due to the presence of phenol-rich phlorotannins that only occur in brown seaweeds [40,107]. A similar phenol content was reported by Nwosu et al. [41] from seaweeds harvested in Ireland, extracted either with methanol or acetonitrile, followed by solid phase extraction (SPE). *A. esculenta* ranged from 28.6 ± 0.4 to 114.6 ± 2.5 µg PE/g, *U. lactuca* 28.6 ± 0.4 to 90.25 µg PE/g, and *P. palmata* from 9.5 ± 0.2 to 205.8 ± 3.7 µg PE/g. In all cases, the total phenol content (TPC) was greatest in the acetonitrile/SPE extracts. Other published values for TPC are generally considerably higher than in the present study, particularly for *Alaria* sp. For example, Castejón et al. [74] reported the TPC of Icelandic *A. esculenta* in milligram gallic acid equivalents (GAE) as 9.37 ± 4.07 mg GAE/g, 1.59 ± 9.58 mg GAE/g (*U. lactuca*), and 1.81 ± 1.04 mg GAE/g (*P. palmata*) using pulsed electric field extraction. Polyphenols were extracted from Norwegian seaweeds by Roleda et al. [108] using acetone/water 80/20 (*v*/*v*). TPC ranged from 14 to 61 mg PE/g in *A. esculenta* and 2 to 6 mg PE/g in *P. palmata*, being significantly higher in spring-harvested biomass than summer and autumn. The different extraction methods as well as spatial and seasonal variations may account for the differing TPC values compared to the present study.

The total lipid contents were all below 1%, ranging from 0.18 ± 0.02% to 0.27 ± 0.09% and 0.48 ± 0.07% in *P*. *palmata*, *A. esculenta*, and *U. lactuca*, respectively. Seaweeds (macroalgae) predominantly have very low lipid contents compared to microalgae [109]. Mæhre et al. [67] found slightly higher lipid contents in *A. esculenta* (1.30 ± 0.05%), *U. lactuca* (2.62 ± 0.14%), and *P*. *palmata* (1.33 ± 0.05%) harvested during May and June in Norway, as did Foseid et al. [110] of 1.1 ± 0.1% in *A. esculenta* and 2.8 ± 0.1% in *P*. *palmata*.

Although comparable compositional contents have previously been published for the three seaweeds in the present study, both lower and higher values also appear in the literature. The impact of seasonal variations, geographic location, biotic or abiotic stresses, and storage conditions may account for the variations in protein, ash, carbohydrate, and lipid content in wild harvested seaweeds. In addition, some published protein values were calculated using nitrogen conversion factors of 6.25 or 5, which can result in an overestimation of the protein content compared to the sum of proteomic amino acids [50], as opposed to the lower factors of 4.45, 4.10, and 4.15 used in the present study.

### 4.2. Fatty Acid Profile

Despite the low lipid contents in seaweeds, the fatty acids they contain are predominantly polyunsaturated including the essential omega-6 fatty acids ALA and LNA. Dietary LNA and ALA are essential for humans and must be consumed in the diet [64]. These are converted in the body to omega-3 EPA and DHA and function as components of the myelin sheath of nerves, inflammatory responses, and molecules involved in regulating blood pressure [111,112]. Algae are unusual amongst plants in that they are the only non-animal, plant source of omega-3 EPA and DHA [113]. The WHO recommends an omega-6 fatty acid intake of 2.5–9% and an omega-3 fatty acid intake of 0.5–2% of total calorific energy/day [114].

In the present study, the total FA concentration ranged from 15.496 mg FA/g lipid extract in *U. Lactuca* to 19.372 mg FA/g in *P. palmata*, and was the greatest in *A. esculenta* (22.322 mg FA/g) (Table 4). C16:0 palmitic acid was the most prevalent SFA in all seaweeds, being highest in *P. palmata* (10.228 mg FA/g). The second most prevalent was C14:0 myristic acid, followed by C22:0 behenic acid. MUFAs were observed in the following order of abundance in all seaweeds: C18:1 trans-9-elaidic acid, cis C18:1 (n-9) oleic acid, and C16:1 palmitoleic acid. Total MUFAs were greatest in *U. lactuca* (7.256 mg FA/g). *A. esculenta* had the highest total PUFA content (10.583 mg FA/g), followed by *P. palmata* (7.848 mg FA/g) and *U. lactuca* (4.149 mg FA/g). The ALA content was 2.009 mg FA/g (*A. esculenta*), 0.401 mg FA/g (*P. palmata*), and 0.240 mg FA/g (*U. lactuca*). The LNA content was also highest in *A. esculenta* (1.507 mg FA/g), followed by 0.650 mg FA/g in *U. lactuca* and 0.597 in *P. palmata*. The most prevalent PUFA in all three seaweeds was EPA, which was 70.89%, 54.88%, and 11.25% of the total PUFA in *P. palmata*, *U. Lactuca,* and *A. esculenta*, respectively. *U. lactuca* contained the most DHA (1.240 mg FA/g), followed by *P. palmata* (0.354 mg FA/g) and *A. esculenta* (0.115 mg FA/g). These high EPA and DHA contents add to the nutritional benefit of consuming seaweeds.

Both n-6 and n-3 fatty acids are essential, however, consuming them in an imbalanced ratio can result in chronic inflammatory diseases such as obesity, rheumatoid arthritis, non-alcoholic fatty liver, and cardiovascular disease [115]. The ratio of n-6:n-3 consumption in developed countries has risen to approximately 20:1 in the last two decades [116]. A ratio between 2.5:1 and 4:1 (n-6:n-3) is generally recommended to prevent chronic diseases associated with excess n-6 monounsaturated fat consumption [117]. In the present study, the n-6:n-3 ratio in *A. esculenta* was almost equal (1.10:1.02), while *U. lactuca* had approximately three times more n-3 than n-6 (1.01:3.13). The n-3 content of *P. palmata* was more than sixfold greater than its n-6 content (1.15:6.70).

Previously, Pereira et al. [118] reported the total FA contents of Portuguese seaweeds harvested in May to be highest in brown species, followed by red, then green, in accordance with the present study. Foseid et al. [110], [112] and Mæhre et al. [67] found similar distributions of total FA, SFA, MUFA, PUFA, and n-6:n-3 ratios in Norwegian summer harvested *A. esculenta*, *U. lactuca,* and *P. palmata*.

Although seaweeds are good sources of the essential fatty acids ALA and LNA as well as EPA and DHA, the minimal lipid content of the seaweeds in the present study may make them unsuitable for development as omega 3 and 6 nutraceuticals compared to microalgal or fish sources.

### 4.3. Outcome of Bioassays

#### 4.3.1. ACE-1 Inhibition

Within the RAAS, ACE-1, ACE-2, and renin are the primary enzymes involved in the regulation of blood pressure [119]. ACE-1 is a dipeptidyl carboxypeptidase found on the luminal surface of vascular endothelial cells in humans and other mammals [120]. ACE-1 is converted to the active vasoconstrictor ACE-2, which regulates systemic vascular resistance and blood volume [10,11]. The present study found that the 1.0 kDa Pep fractions of *P. palmata* had an ACE-1 inhibitory activity equivalent to the control, captopril©, while neither *A. esculenta* nor *U. lactuca* Pep, or any of the Psac and Pphen extracts, were significantly effective. ACE-1 inhibition ranged from most effective by *P. palmata* Pep (94.29 ± 3.07 µg/mL) to least effective (704.69 ± 20.72 µg/mL) by *A. esculenta* Psac. The interaction mechanisms and structure–activity relationship of algal-derived ACE-1 inhibitory peptides have been elucidated in some molecular docking studies. Most have been determined to inhibit ACE-1 in a non-competitive binding mode [37,38,121,122,123].

Similar ranges of inhibition have been previously reported for the *Alaria*, *Ulva*, and *Palmaria* species. In accordance with the present study. Paiva et al. [124] found that bromelain hydrolysates isolated from *Ulva compressa* and *Ulva rigida* had ACE-1 inhibitory activities of 59.80% and 65.68%, respectively (at 1 mg/mL). In terms of IC_50_ values, both lower and higher values for ACE-1 have been previously determined. For example, IC_50_ values in the range of 0.19 to 0.78 mg/mL were reported by Harnedy and FitzGerald [125] from *P. palmata* hydrolysates in vitro. The *P. palmata* was harvested in October, as opposed to in summer, in the present study. In addition, the proteases alcalase and corolase were used instead of papain to prepare the protein hydrolysate fractions [125]. These different enzymatic methods and harvesting dates may have influenced the bio-functional activity of the hydrolysates. Other studies have reported ACE-1 inhibition by *Alaria, Ulva*, and *Palmaria* species and are detailed in Table 6.

#### 4.3.2. α-Amylase Inhibition

Inhibition of the digestive enzyme α-amylase is one therapeutic approach in lowering postprandial blood glucose levels to manage T2D. The α-amylase inhibition IC_50_ values ranged from 147.04 ± 9.72 µg/mL (*A. esculenta* Pphen) to the least effective (441.32 ± 23.51 µg/mL) for *U. lactuca* Pep. Among the three seaweeds, *A. esculenta* Pphen had a significantly greater inhibitory activity against α-amylase than *U. lactuca* or *P. palmata* Pphen. This may be due to the presence of phlorotannins, which are polyphenols unique to brown seaweeds [126]. Bromophenols, which occur predominantly in red and green seaweeds, can also inhibit α-amylase as well as the enzyme aldose reductase, which converts glucose to sorbitol in the polyol pathway and has been used in the prevention of diabetic complications [127]. Polyphenols derived from the *Alaria, Ulva*, and *Palmaria* species have previously exhibited α-amylase effects in vitro and in vivo. For example, Song and Wang [128] supplemented the diet of high-fat-diet-treated mice with an ethanol extracted polyphenol of *Ulva prolifera*. After 8 weeks, there was a significant improvement in glucose tolerance and insulin resistance, and a reduction in the high-fat-diet-induced weight gain of liver and fat in the *U. prolifera* supplemented mice. IC_50_ values ranging from 0.90 to 1.40 µg/mL in *A. esculenta* and *P. palmata* polyphenol extracts were reported by Nwosu et al. [41] for α-amylase inhibition.

Polysaccharides such as fucoidan, laminarin, alginate, ulvan, and porphyran are unique to seaweeds and have different bioactive properties than the polysaccharides found in terrestrial plants [129]. An in silico grid-based docking study by Lakshmanasenthil et al. [46] found that fucoidan was a potent inhibitor of α-amylase as well as the T2D-related enzyme, α-glucosidase, based on number of interactions, hydrogen bond length, and binding energy. Fucoidan has also been utilised in combination therapy, where a reduced dose of a pharmaceutical inhibitor, such as acarbose, is combined with a naturally occurring extract. For instance, Mabate et al. [130] inhibited α-amylase and α-glucosidase in vitro using fucoidan in conjunction with acarbose. Fucoidan inhibited α-glucosidase (IC_50_ 19 µg/mL) more effectively than acarbose (IC_50_ 332 µg/mL) but had no significant effect on α-amylase. Conversely, acarbose did effectively inhibit α-amylase (IC_50_ 109 µg/mL). Combining fucoidan and acarbose actually had a synergistic effect, and increased the α-glucosidase inhibitory activity significantly (>70%) compared to acarbose alone.

Peptides have been less widely reported in the literature as inhibitors of α-amylase, however, Admassu et al. [39] found that two peptides, (Gly-Gly-Ser-Lys) and (Glu-Leu-Ser), from the red seaweed *Porphyra* sp. had IC_50_ values of 2.58 ± 0.08 mM and 2.62 ± 0.05 mM, respectively. Inhibitory kinetics showed that the peptides exhibited a non-competitive binding mode with α-amylase. Further studies on α-amylase inhibition by *Alaria, Ulva*, and *Palmaria* species are summarised in Table 6.

#### 4.3.3. Lipase Inhibition

Adipose (lipid) tissues are required for several bodily functions including insulation, the storage of free fatty acids for later caloric need, and the absorption of fat-soluble vitamins [131]. However, an excess accumulation of fat increases an individual’s risk for heart disease, MS, and other disorders [132]. The same pattern observed in α-amylase inhibition occurred with lipase, where *A. esculenta* Pphen had the greatest inhibitory efficacy (IC_50_ value 106.21 ± 6.53 µg/mL), followed by the Psac extracts, with peptides having the least efficacy (*A. esculenta* Pep 653.31 ± 36.44 µg/mL). The superior inhibition by *A. esculenta* may be due to its TPC (82.83 µg PE/g) being greater than *U. lactuca* or *P. palmata* as well as its previously reported high alginate content of up to 40% [133]. Molecular docking studies have shown that the carboxyl groups of phlorotannins, polyphenols, and polysaccharides can donate a proton to the amino acids histidine and serine at the active site of lipase, preventing them from forming a complex with its triglyceride substrate [23,134,135].

Previously, Nakayama and Shimada [136] ameliorated several anti-obesity traits in mice and zebrafish including dyslipidaemia, hepatic steatosis, and visceral adiposity by including *Palmaria mollis* (2.5% DW) in feed for 4 weeks. Further studies on lipase inhibition by *Alaria, Ulva*, and *Palmaria* species are summarised in Table 6.

**Table 6 foods-14-00284-t006:** Previously published ACE-1, α-amylase, and lipase inhibition by *Alaria, Ulva*, and *Palmaria* species.

Enzyme Inhibited	Seaweed Species	Extract Type	% Inhibition or Effect	IC_50_ Value	Study Type	Reference
	*A. esculenta*	Alcalase peptide hydrolysate	48% (5 mg/mL)	/	In vitro	Sapatinha et al. [38]
ACE-1	*U. rigida*	Pepsin and bromelain peptide hydrolysates	/	483 µg/mL (whole hydrolysate) 95 µg/mL (1 kDa MWCO permeate)	In vitro	Paiva et al. [137]
	*U. intestinalis*	Trypsin peptide hydrolysates (FGMPLDR and MELVLR)	/	183 µg/mL 179 µg/mL	In vitro	Sun et al. [16]
	*U. lactuca*	Papain protein hydrolysates	Whole hydrolysate 82.37%, 1 kDa 93.03%, 3 kDa 86.64% (all 1 mg/mL)	/	In vitro	Garcia-Vaquero et al. [68]
	*P. palmata*	Pepsin, trypsin and chymotrypsin peptide hydrolysates (LRY, VYRT, FEQDWAS and LDY)	88% (5 mg/mL)	LRY 0.044 μmol VYRT 0.14 μmol FEQDWAS 2.80 μmol LDY 6.10 μmol	In vitro	Furuta et al. [31]
	*P. palmata*	Oral, gastric and duodenal enzymatically digested protein hydrolysate	32.6 ± 1.2% (3 mg/mL)	/	In vitro	Vasconcelos et al. [35]
	*P. palmata*	Chymotrypsin peptide hydrolysate, <10 kDa	67.74 ± 3.91% (5 mg/mL)	/	In vitro	Beaulieu et al. [34]
	*A. esculenta*	Polyphenol extracts	/	0.90 µg/mL	In vitro	Nwosu et al. [41]
α-Amylase	*U. lactuca*	Water and ethanol extracted polysaccharides	↓ α-amylase in blood plasma (53%) and small intestine (34%) ↓ blood glucose concentration (43%)	/	In vivo trial with diabetic rats (*n* = 40) 30 d.	BelHadj et al. [20]
	*U. prolifera*	Polysaccharide	↑ glucose tolerance and insulin resistance ↓ high-fat-diet-induced weight gain of liver and fat	/	In vivo trial with high-fat-diet-treated male mice (*n* = 32), 8 wk	Song and Wang [128]
	*P. palmata*	Polyphenol extracts	/	1.40 µg/mL	In vitro	Nwosu et al. [41]
Lipase	*U. australis*	Dichloromethane extract of polyphenols	66.27 ± 2.01% (10 mg/mL)	6.65 mg/mL	In vitro	Trenti et al. [138]
	*U. prolifera*	Sulphated polysaccharide	93.73 ± 0.92% (8 mg/mL)	/	In vitro	Yuan et al. [139]
	*U. lactuca*	Water and ethanol extracted polysaccharides	Lipase activity in blood plasma (↓235%) and small intestine (↓287%)	/	In vivo trial with diabetic rats (*n* = 40) 30 d.	BelHadj et al. [20]
	*U. lactuca*	Viscozyme® and Flavourzyme® extracted polysaccharide	/	50 mg/mL	In vitro	Tong et al. [45]

## 5. Conclusions

The key findings of this study indicate that *P. palmata* peptides, *A. esculenta* polyphenols, and *U. lactuca* polysaccharides had an equal or greater efficacy in vitro than the three positive control pharmaceutical inhibitors. Potential applications may involve the development of the extracts as anti-hypertensive, anti-diabetic, and anti-obesity supplements or as human or animal functional food ingredients to support metabolic health. Limitations of the present study include a lack of cytotoxicity assessment or characterisation of the bioactive molecules using LC-MS or NMR. Recommendations for future research include amino acid sequencing of the peptide hydrolysates and molecular characterisation of the polyphenols and polysaccharides after purification and standardisation of the extracts. The effect of digestion on the bioavailability and bioactivity should be assessed using simulated in vitro models. Micro-encapsulation may enhance the bioavailability and bioaccessibility of these extracts, particularly the oxygen and light sensitive polyphenols. Cell viability and toxicity assessment is required in cell models, followed by carefully planned animal trials and human intervention studies.

The data from this study may inform future studies using these three seaweeds for functional food use as anti-hypertensive, anti-diabetic, and weight control ingredients in the strategy of metabolic syndrome management.

## Figures and Tables

**Figure 1 foods-14-00284-f001:**
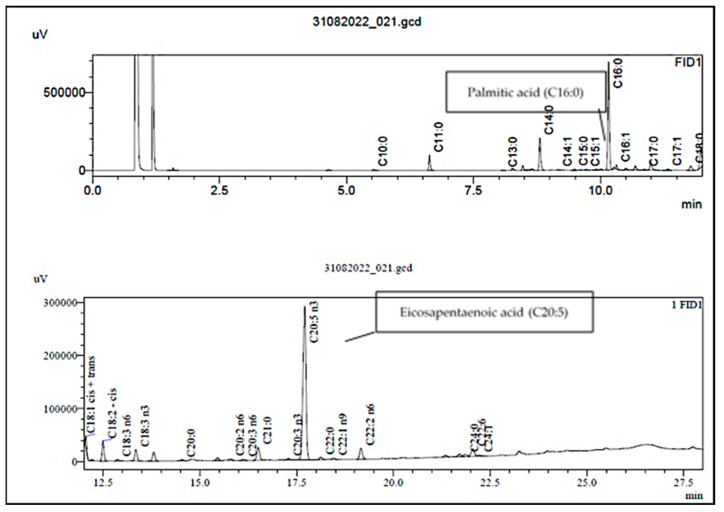
GC chromatogram of *P. palmata* FAME, showing palmitic acid and eicosapentaenoic acid (EPA) as the prominent peaks.

**Figure 2 foods-14-00284-f002:**
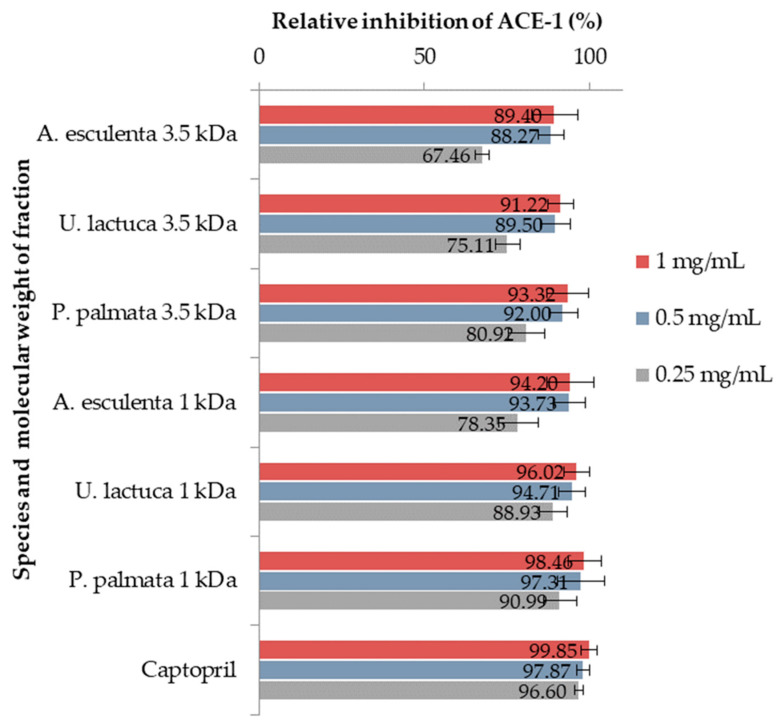
Percentage inhibition of ACE-1 by 1 kDa and 3.5 kDa peptide hydrolysate fractions or captopril© at concentrations of 0.25, 0.5, and 1.0 mg/mL. All analyses were conducted in triplicate and expressed as the means of three ± standard deviation.

**Table 1 foods-14-00284-t001:** Proximate composition of whole seaweeds versus the crude protein, polysaccharide, and polyphenol extracts.

	Whole Seaweeds			After Extraction			
	*A. esculenta*	*U. lactuca*	*P. palmata*		*A. esculenta*	*U. lactuca*	*P. palmata*
	% of whole seaweed (DW)				% of crude protein extract (DW)		
Protein	10.31 ± 0.13	6.61 ± 0.08	9.95 ± 0.21	Protein	74.34 ± 4.56	78.77 ± 3.81	60.52 ± 3.09
Moisture	87.79 ± 4.65	87.60 ± 2.53	91.36 ± 8.38				
Ash	16.10 ± 0.08	14.57 ± 0.37	10.11 ± 0.08				
Total lipids	0.27 ± 0.09	0.48 ± 0.07	0.18 ± 0.02		% of polysaccharide extract (DW)		
Total carbohydrates	50.49 ± 2.44	66.04 ± 3.67	69.77 ± 4.89	Total carbohydrates	86.98 ± 3.66	90.57 ± 4.28	91.98 ± 4.04
Total fibre	38.10 ± 1.50	37.50 ± 1.04	36.10 ± 1.67	Total fibre	40.30 ± 2.84	42.10 ± 2.23	56.00 ± 3.70
Soluble fibre	7.40 ± 0.30	17.10 ± 0.42	6.70 ± 0.21	Soluble fibre	38.70 ± 2.35	41.10 ± 2.64	53.60 ± 3.52
Insoluble fibre	30.50 ± 1.48	20.40 ± 1.20	29.40 ± 1.33	Insoluble fibre	1.60 ± 0.06	1.00 ± 0.00	2.40 ± 0.09
					µg PE/g polyphenol extract (DW)		
				Total phenolics	82.83 ± 4.99	75.17 ± 3.14	69.33 ± 2.94

**Table 2 foods-14-00284-t002:** Degree of hydrolysis of the papain hydrolysates (mean, *n* = 3, ±SD).

Degree of Hydrolysis (%)		
*A. esculenta*	*U. lactuca*	*P. palmata*
71.43 ± 2.79	75.06 ± 4.03	80.62 ± 4.86

**Table 3 foods-14-00284-t003:** Monosaccharide contents of the polysaccharide extracts (mean, *n* = 3, ±SD).

	g/100 g Psac Extract (DW)		
	*A. esculenta*	*U. lactuca*	*P. palmata*
L-rhamnose	2.47 ± 0.08	4.98 ± 0.09	2.76 ± 0.13
D-galactose	0.45 ± 0.01	0.74 ± 0.14	2.95 ± 0.31
L-arabinose	4.60 ± 0.26	4.37 ± 0.42	7.90 ± 0.62
D-glucose	4.16 ± 0.29	1.30 ± 0.19	0.12 ± 0.02
D-fructose	0.00 ± 0.00	0.35 ± 0.07	0.00 ± 0.00
D-mannose	0.00 ± 0.00	0.00 ± 0.00	0.00 ± 0.00
∑ Monosaccharides	11.68	11.74	13.75

**Table 4 foods-14-00284-t004:** Fatty acid methyl ester profile of the seaweed lipid extracts.

	mg FA/g Lipid Extract (DW)		
	*A. esculenta*	*U. lactuca*	*P. palmata*
C4:0 Butyric acid	0.000	0.000	0.000
C6:0 Caproic acid	0.000	0.000	0.000
C8:0 Caprylic acid	0.000	0.000	0.000
C10:0 Capric acid	0.179	0.000	0.215
C11:0 Undecanoic acid	0.000	0.000	0.820
C12:0 Lauric acid	0.000	0.000	0.000
C13:0 Tridecanoic acid	0.166	0.000	0.091
C14:0 Myristic acid	2.010	0.860	1.755
C15:0 Pentadecanoic acid	0.155	0.130	0.130
C16:0 Palmitic acid	4.129	1.140	5.346
C17:0 Heptadecanoic acid	0.125	0.120	0.114
C18:0 Stearic acid	0.358	0.300	0.440
C20:0 Arachidic acid	0.225	0.000	0.211
C21:0 Heneicosanoic acid	0.000	0.140	0.434
C22:0 Behenic acid	0.000	1.270	0.241
C23:0 Tricosanoic acid	0.000	0.140	0.191
C24:0 Lignoceric acid	0.000	0.000	0.239
∑ SFA	7.347	4.092	10.228
C14:1 Myristoleic acid	0.116	0.100	0.097
C15:1 cis-10-Pentadecenoic acid	0.107	0.100	0.112
C16:1 Palmitoleic acid	0.571	2.230	0.251
C17:1 cis-Heptadecenoic acid	0.244	0.120	0.160
C18:1 trans-9-Elaidic acid and cis C18:1 (n-9) Oleic acid	3.242	1.240	0.434
C20:1 (n-9) cis-11-Eicosenoic acid	0.000	3.330	0.000
C22:1 (n-9) Erucic acid	0.000	0.000	0.127
C24:1 Nervonic acid	0.110	0.140	0.115
∑ MUFA	4.392	7.256	1.296
C18:2 (n-6) cis-Linoleic acid (LNA)	1.507	0.650	0.597
C18:2 (n-6) trans-Linolelaidic acid	0.000	0.100	0.000
C18:3 (n-6) gamma-Linolenic acid	0.276	0.120	0.122
C18:3 (n-3) alpha-Linolenic acid (ALA)	2.009	0.240	0.401
C20:2 (n-6) cis-11,14-Eicosadienoic acid	0.224	0.140	0.122
C20:3 (n-6)–cis-8,11,14-Eicosatrienoic acid (dihomo-gamma-linolenic acid)	0.175	0.000	0.117
C20:4 (n-6) Arachidonic acid	3.449	0.000	0.000
C20:3 (n-3) cis-11,14,17-Eicosatrienoic acid	0.175	0.210	0.131
C20:5 (n-3) cis-5,8,11,14,17-Eicosapentaenoic acid (EPA)	2.277	1.190	5.563
C22:2 (n-6) cis-13,16-Docasadienoic acid	0.377	0.260	0.441
C22:6 (n-3) cis-4,7,10,13,16,19-Docosahexaenoic acid (DHA)	0.115	1.240	0.354
∑ PUFA	10.583	4.149	7.848
∑ n-6	5.508	1.010	1.154
∑ n-3	5.075	3.139	6.694
∑ n-6:∑ n-3 ratio	1.10:1.02	1.01:3.13	1.15:6.70
Total mg FA/g lipid extract	22.322	15.496	19.372

**Table 5 foods-14-00284-t005:** Half-maximal inhibitory concentration (IC_50_) of the seaweed extracts and control inhibitors on ACE-1, α-amylase, and lipase activity.

		Enzyme IC_50_ (µg/mL) Values		
Species and Extract		ACE-1	α-Amylase	Lipase
*A. esculenta*	Psac	704.69 ± 20.72 ^i^	265.49 ± 18.36 ^f^	288.35 ± 11.49 ^e^
	Pphen	365.86 ± 11.53 ^d^	147.04 ± 9.72 ^a^	106.21 ± 6.53 ^a^
	Pep (1 kDa)	122.38 ± 9.11 ^c^	374.20 ± 20.75 ^h^	653.31 ± 36.44 ^h^
*U. lactuca*	Psac	551.73 ± 14.29 ^g^	168.06 ± 10.53 ^b^	151.26 ± 9.04 ^c^
	Pphen	436.80 ± 20.63 ^f^	203.55 ± 12.77 ^d^	158.89 ± 8.71 ^c^
	Pep (1 kDa)	108.69 ± 8.32 ^b^	441.32 ± 23.51 ^i^	606.90 ± 30.12 ^h^
*P. palmata*	Psac	646.52 ± 18.29 ^h^	215.80 ± 11.75 ^de^	374.78 ± 21.92 ^f^
	Pphen	398.10 ± 17.49 ^e^	278.66 ± 13.79 ^f^	167.80 ± 9.36 ^d^
	Pep (1 kDa)	94.29 ± 3.07 ^a^	306.75 ± 19.42 ^g^	425.53 ± 20.47 ^g^
	Captopril©	91.83 ± 2.68 ^a^	N/A	N/A
Controls	Acarbose	N/A	185.67 ± 12.48 ^c^	N/A
	Orlistat	N/A	N/A	139.74 ± 9.33 ^b^

All analyses were conducted in triplicate and expressed as the means ± standard deviation. One-way ANOVA and Tukey’s HSD post hoc test were used to determine statistically significant differences (*p* ≤ 0.05) between the means of each column, indicated by different letters.

## Data Availability

The original contributions presented in the study are included in the article, further inquiries can be directed to the corresponding author.

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
