# Peer review of "Alaria esculenta, Ulva lactuca, and Palmaria palmata as Potential Functional Food Ingredients for the Management of Metabolic Syndrome"

_foods, 2025, doi:10.3390/foods14020284_

Round 1
Reviewer 1 Report
Comments and Suggestions for Authors
Review Report Foods "Bioactives from Alaria esculenta, Ulva lactuca, and Palmaria palmata as Potential Functional Food Ingredients to Prevent Metabolic Syndrome”.
The study investigates bioactive compounds extracted from three edible Irish seaweeds (Alaria esculenta, Ulva lactuca, and Palmaria palmata) to assess their potential as functional food ingredients in managing metabolic syndrome. Specifically, the study focuses on the inhibitory effects of seaweed-derived polysaccharides, polyphenols, and peptides on key enzymes associated with hypertension (ACE-1), type 2 diabetes (α-Amylase), and obesity (lipase).
General comments and suggestions:
Abstract is a bit confusing and should be more focused and simplified. Introduction is comprehensive and well organized.
The study effectively examines a range of bioactive components, however only lipid profile was characterized. The potency of the study will be highest if other biomolecule profiles are evaluated.
Seaweed bioactives are compared with recognized positive control drugs (Captopril for ACE-1, Acarbose for α-Amylase, and Orlistat for lipase). This comparison enhances the relevance of findings, showing that some extracts, especially from P. palmata, are nearly as effective or more effective than pharmaceutical counterparts.
The study uses in vitro enzyme inhibition assays (ACE-1, α-Amylase, and lipase) directly targets key biomarkers of metabolic syndrome. This approach provides a mechanistic basis for the potential benefits of these bioactive compounds.
Issues:
It is not clear how many samples was analysed for each algae species. Three replicates, but how many samples were analysed? Only one?
May the sampling region and season affect the composition of the matrices?
Absence of Toxicological Evaluation: Although the study assesses potential health benefits, it does not examine the safety profile of the extracts at the tested concentrations. Preliminary toxicological assessments should be carried out by simple cytotoxicity tests using simple models, for example on Caco-2 cells. I suggest authors consider adding this experimental part.
Conclusions section should be developed to include a summary of key findings, potential applications, and recommendations for future research would enhance the study impact. I suggest authors highlight also future perspectives and research objectives.
Based on these comments I recommend major revisions, underlining that this manuscript is a good candidate for the editorial process in Foods.
Author Response
Please see the response to reviewer's comments:
|
No. |
Comment |
Response |
|
1 |
Abstract is a bit confusing and should be more focused and simplified. Introduction is comprehensive and well organized. |
The Abstract has been simplified and made more concise. |
|
2 |
It is not clear how many samples was analysed for each algae species. Three replicates, but how many samples were analysed? Only one? |
Due to the extensive range of assays and analyses undertaken in the study, it was only feasible to select one biomass sample from each class of seaweed (red, brown, and green), analysed in triplicate. |
|
3 |
May the sampling region and season affect the composition of the matrices? |
Seasonal and geographical variations may affect the composition of the seaweeds and will be included in future study plans for these species. |
|
4 |
Absence of Toxicological Evaluation: Although the study assesses potential health benefits, it does not examine the safety profile of the extracts at the tested concentrations. Preliminary toxicological assessments should be carried out by simple cytotoxicity tests using simple models, for example on Caco-2 cells. I suggest authors consider adding this experimental part. |
Toxicological assessment was not within the scope of the present study, which aimed to screen seaweeds for their enzyme inhibitory activity. Further assessment of the extracts would entail cytotoxicity testing in cell models. |
|
5 |
Conclusions section should be developed to include a summary of key findings, potential applications, and recommendations for future research would enhance the study impact. I suggest authors highlight also future perspectives and research objectives. |
The Conclusion section has been revised to include the key findings and future perspectives. |
|
6 |
Based on these comments I recommend major revisions, underlining that this manuscript is a good candidate for the editorial process in Foods. |
All comments and recommendations have been incorporated into the revised manuscript. |
Reviewer 2 Report
Comments and Suggestions for Authors
This study investigated the inhibitory activities of polysaccharides, polyphenols, and peptide compounds extracted from edible seaweed such as Alaria esculenta, Ulva lactuca, and Palmaria palmata on angiotensin-converting enzyme (ACE-1), α- amylase, and lipase. The results of in vitro enzyme inhibition experiments indicate that these seaweed extracts have the potential to lower blood pressure, blood glucose levels, and absorb excessive fat. But there are still some problems and shortcomings that need to be revised and improved by the author.
1. In this manuscript, Alaria esculenta, Ulva lactuca, and Palmaria palmata were selected as raw materials. Why were these algae selected? Is the selection of these algae representative of other algae?
2. In section 2.3.2, time units are mixed, please unify the format. There are many such cases in the article, please revise the whole text
3. In section 2.3.4, “60 minutes” is changed to “60 min”. There are many such cases in the article, please revise the whole text
4. In section 2.3.9, please check whether "1.h" is correct.
5. The data in the manuscript lacked significance analysis, and a statistical significance test was recommended.
6. Many references are cited in the discussion section. Do these references support what the author is trying to express?
7. The author cited the application of a large number of nutrients in seaweed in food additives, but there was no relevant research in this study.
8. The format of references in Table 6 is not uniform, please modify it.
9. This manuscript lacks innovation. What is the innovation between it and previous research?
Author Response
Please see the response to reviewer's comments:
|
No. |
Comment |
Response |
|
1 |
In this manuscript, Alaria esculenta, Ulva lactuca, and Palmaria palmata were selected as raw materials. Why were these algae selected? Is the selection of these algae representative of other algae? |
These seaweeds were selected as they are three of the most commonly consumed algae within each class in Ireland. A more comprehensive range of species was not feasible in the present study due to the extensive range and number of assays undertaken, but will be included in plans for future studies. |
|
2 |
In section 2.3.2, time units are mixed, please unify the format. There are many such cases in the article, please revise the whole text. |
‘hr’ has been corrected to ‘h’ throughout. |
|
3 |
In section 2.3.4, “60 minutes” is changed to “60 min”. There are many such cases in the article, please revise the whole text. |
“minutes” has been corrected to ‘min’ throughout. |
|
4 |
In section 2.3.9, please check whether "1.h" is correct. |
"1.h" could not be located in section 2.3.9, or throughout the manuscript. |
|
5 |
The data in the manuscript lacked significance analysis, and a statistical significance test was recommended. |
The significance of data in each Table was analysed only when a comparison was being made amongst the means, as set out in section 2.3.1 Statistical analysis. Means that were statistically indistinguishable were assigned the same subgroups (a, b, c, ab). For example, in Table 5. Half-maximal inhibitory concentration (IC50) of seaweed extracts and control inhibitors on ACE-1, α-Amylase, and lipase activity. Analysis of variance was not applied to Tables 1, 3, 4, or Fig. 2 as these data sets intend only to show a quantity of something that was measured in each individual seaweed, or to determine which of three had the highest percentage. It was not expected that there would be any significance or correlation between their means. |
|
6 |
Many references are cited in the discussion section. Do these references support what the author is trying to express? |
References 66 to 139 show the context of the results in terms of similar studies and support the relevance of algal components as functional foods that have shown positive effects in the treatment of metabolic syndrome in animal studies and in vitro. |
|
7 |
The author cited the application of a large number of nutrients in seaweed in food additives, but there was no relevant research in this study. |
The scope of the present study was limited to in vitro benchtop assays and proximate analysis. It was not possible to conduct any analyses of the extracts as a food ingredient in animal models or humans. |
|
8 |
The format of references in Table 6 is not uniform, please modify it. |
Three references have been corrected. |
|
9 |
This manuscript lacks innovation. What is the innovation between it and previous research? |
We respectfully disagree with this comment from the reviewer. The manuscript is very innovative as it describes, for the first time, inhibition by selected extracts of polyphenols, peptides and polysaccharides from these algae against the physiologically relevant enzymes alpha-amylase, lipase and ACE-1. From the literature, inhibitors of these enzymes were identified and are discussed in the paper but they are from different seaweeds and none of the previously described papers have shown inhibition of enzymes at these physiologically relevant levels against these specific enzymes. |
Reviewer 3 Report
Comments and Suggestions for Authors
The present manuscript evaluated nutritional values of three species of Irish seaweed (Alaria esculenta, Ulva lactuca and Palmaria palmata) and their activities including ACE-1 inhibition, lipase inhibition and α-Amylase inhibition. It is very interesting, and easy to read, highlighting that these seaweeds show the potential in the management of diseases associated with MS and as foods. However, the following work is required before the manuscript can be considered for publication.
1. The creativity of this work is relatively poor. In the introduction section, as the authors indicated some studies on ACE-1, lipase and α-Amylase inhibition by seaweed extracts, have been reported. What is the innovation for present study? In addition, as the authors have indicated, the main purpose of this investigation was to extract Psac, Phen and Pep from the seaweeds and screen the extracts for their inhibitory effects against ACE-1, lipase and α-Amylase. However, in the results and discussion section, the authors are more focused on their study on proximate analysis of the seaweeds. Furthermore, the removal of the last paragraph in the introduction section was suggested. In word, the authors should revise the introduction section carefully.
2. In the result section (3.3.1 ACE-1 percentage inhibition by peptide hydrolysates), the authors found that the 1kDa fractions of peptide hydrolysates exhibited greater efficacy in ACE-1 inhibition. In the subsequent studies, the same fractions were also employed to investigate the lipase and α-Amylase inhibition. It is unclear how the authors reached the conclusion that the 1kDa fractions also demonstrated effective inhibition of lipase and α-Amylase.
3.Table 1 should be revised carefully. It is not easy to read; In “3.2. Fatty acid methyl ester content”, the FA content of each seaweed’s lipid extract as quantified by gas chromatography is set out in Table 5. Here, the discussion for table 5 should be relocated behind table 4; Additionally, it was advise to delete table 6. It is unclear what is the information that the authors are planning to provide to the readers?
Author Response
Please see the response to reviewer's comments:
|
No. |
Comment |
Response |
|
1 |
The creativity of this work is relatively poor. In the introduction section, as the authors indicated some studies on ACE-1, lipase and α-Amylase inhibition by seaweed extracts, have been reported. What is the innovation for present study? In addition, as the authors have indicated, the main purpose of this investigation was to extract Psac, Phen and Pep from the seaweeds and screen the extracts for their inhibitory effects against ACE-1, lipase and α-Amylase. However, in the results and discussion section, the authors are more focused on their study on proximate analysis of the seaweeds. Furthermore, the removal of the last paragraph in the introduction section was suggested. In word, the authors should revise the introduction section carefully |
Innovative aspects of the present study are now highlighted on page 3 in the following section “This is the first time, to the best of the authors knowledge, that all extracts (Pep, Pphen and Psac) from these particular seaweeds were screened for their ability to inhibit lipase, α-amylase and ACE-1. Inhibition of these enzymes using these specific fractions is innovative as it enhances the inhibition potential compared to whole seaweeds and additionally makes it easier for the actives to be formulated at physiologically relevant concentrations into final food products without impacting sensory acceptability or compromising bioactivity.
Proximate analysis of the seaweeds was conducted to assess the differences in total protein, polysaccharides, and fibre contents of whole seaweed biomass versus the extracts.
The Introduction section has been revised in Word. We also discuss the extracts and their bioactive potential now in the results and discussion section in the text from pages 20-24 (4 pages dedicated to the bioactivity aspect of the work).
|
|
2 |
In the result section (3.3.1 ACE-1 percentage inhibition by peptide hydrolysates), the authors found that the 1kDa fractions of peptide hydrolysates exhibited greater efficacy in ACE-1 inhibition. In the subsequent studies, the same fractions were also employed to investigate the lipase and α-Amylase inhibition. It is unclear how the authors reached the conclusion that the 1kDa fractions also demonstrated effective inhibition of lipase and α-Amylase.
|
We wanted to assess all 1 kDa peptide permeate fractions against the three enzymes. We have revised the text accordingly as follows: “Since the 1 kDa fractions were found to be more effective, these were used to conduct further ACE-1 inhibitory assays at concentrations of 62.5, 125, 250, 500 and 1000 µg/mL to determine half-maximal inhibitory concentrations.”
|
|
3 |
Table 1 should be revised carefully. It is not easy to read.
In “3.2. Fatty acid methyl ester content”, the FA content of each seaweed’s lipid extract as quantified by gas chromatography is set out in Table 5. Here, the discussion for table 5 should be relocated behind table 4.
Additionally, it was advise to delete table 6. It is unclear what is the information that the authors are planning to provide to the readers? |
Table 1 has been revised and restructured.
Table 4. Fatty acid methyl ester profile of seaweed lipid extracts was accidentally referred to as ‘Table 5’ in the first sentence of section 3.2. We apologise for this error. It has been corrected. Table 4 and Fig. 1 ae also now placed directly after their first mention in the text.
Table 6. Shows the potential of algal components as functional foods that have previously shown inhibitory effects against ACE-1, α-Amylase and lipase. |
Round 2
Reviewer 1 Report
Comments and Suggestions for Authors
The work of revision, as well as the limited number of samples analyzed and the variables not considered (It Is not enough to say "will be included in future study planEct.."), really limit the "potency" of this study. For this reasons, in my opinion, in the current form the manuscript is not eligible.
I have to add that if the author are not willing to do additional analyses, experiments and considerations, they should change the title of the manuscript, that is misunderstanding.
Author Response
The work of revision, as well as the limited number of samples analyzed and the variables not considered (It Is not enough to say "will be included in future study planEct.."), really limit the "potency" of this study. For this reasons, in my opinion, in the current form the manuscript is not eligible.
Response:
|
In this study, we aimed to assess the feasibility of deriving functional food ingredients from three commonly consumed seaweeds. In vitro enzyme inhibitory assays were selected to determine whether the seaweed extracts had potential to inhibit enzymes associated with metabolic syndrome, which has become a global health concern. This was a proof-of-concept study to inform the direction of future studies with the seaweeds such as cell toxicity and animal trials. The intention was not to claim that the extracts are ready to use as pharmaceuticals. |
We have included our responses to comments with the last revisions. As explained to the editor, our study was not a seasonality study - it focused on extracts and their impact on enzymes for health.
Reviewer 2 Report
Comments and Suggestions for Authors
All issues were addressed.
Author Response
We thank the reviewer for accepting our response to their queries from round 1.
Reviewer 3 Report
Comments and Suggestions for Authors
The paper is significant improved. However, similar match rate of this manuscript reached to 31%. It is too high, and should be reduced.
Author Response
The paper is significant improved. However, similar match rate of this manuscript reached to 31%. It is too high, and should be reduced.
Response: The manuscript has been revised to reduce the match rate.